# Optimizing Revenue through User Coupon Recommendations in Truthful Online Ad Auctions

## Abstract

Online advertising serves as the primary revenue source for numerous Internet companies, which typically sell advertising slots through auctions. Conventional online ad auctions assume constant click-through rates (CTRs) and conversion rates (CVRs) for ads during the auction process. However, this paper studies a new scenario where advertisers can offer coupons to users, thereby influencing both CTRs and CVRs and consequently, the platform's revenue.

We study how to recommend user coupons to advertisers in truthful auction systems. We model the interaction between the platform and the advertisers as an extensive-form game, where advertisers first report coupon bids to the platform to receive coupon recommendations, and then participate in auctions by reporting their auction bids. Our research identifies a sufficient condition under which the advertisers' optimal strategy is to report their valuations truthfully in both the recommendation and auction stages.

We construct two mechanisms based on these findings. The first mechanism is a distribution-free mechanism, which is easily implementable in industrial systems; and the second is a revenue-optimal mechanism that offers simpler implementation compared to existing work [10]. Both synthetic and industrial experiments show that our mechanisms improve the platform's revenue. Notably, our revenue-optimal mechanism achieves the same outcome compared to existing work by Liu et al. [10], while offering a simpler implementation.

## CCS Concepts

• **Information systems** → **Computational advertising**; • **Theory of computation** → **Algorithmic game theory and mechanism design**.

## Keywords

mechanism design, auction, user coupon recommendation

**ACM Reference Format:**
Anonymous Author(s). 2025. Optimizing Revenue through User Coupon Recommendations in Truthful Online Ad Auctions. In *WWW '25: he Web Conference 2025, April 28-May 2, 2025, Sydney, Australia.* ACM, New York, NY, USA, 9 pages. https://doi.org/10.1145/nnnnnnn.nnnnnnn

## 1 Introduction

Auction design has attracted significant attention in both computer science and economics, leading to the development and implementation of various auction mechanisms across different applications over the past decades [1, 4, 6, 13, 22], including the Vickery-Clarke-Groves (VCG) auction [3, 8, 23], Myerson's optimal auction [15], and the generalized second-price (GSP) auction [6, 22].

One of the most prominent and commercially successful applications of auction theory is online advertising, which contributes to the primary revenue of many Internet companies such as Google, TikTok, and Alibaba. In these systems, in addition to advertisers' bids, relevance scores also play a crucial role in optimizing the auction outcomes. Two widely used relevance scores are the Click-Through Rates (CTRs) and the Conversion Rates (CVRs). The CTR denotes the probability of a user clicking on an advertisement and the CVR represents the probability of a user purchasing the advertiser's product after clicking on the ad.

Consider the example of a per-click auction. Advertisers first submit their bids for a single click. The platform uses the CTRs (and/or the CVRs) to calculate the rank score by multiplying the advertiser's bid with the corresponding relevance score. This rank score is then used to determine both the allocation and the payment.

The CTRs and the CVRs are usually treated as constants in the auction process. However, recent studies [10, 16] start to regard them as variables, considering a setting where the platform can offer coupons to users when displaying the ads. These coupons allow users to purchase products or services at lower prices, making the ads more attractive and therefore increasing both CTRs and CVRs.

In economic theories, the inverse relationship between price and demand is usually described by demand curves [11], where lower prices generally lead to higher demand [17]. Both the CTRs and CVRs can be viewed as proxies for user demand in online advertising. From this perspective, coupons work as a re-pricing mechanism that lowers the prices of products or services in exchange for higher demand. However, advertisers often lack precise knowledge of the demand curves, making it difficult for them to set optimal prices to maximize profit. In contrast, advertising platforms have access to vast amounts of user data, allowing them to train machine learning models to accurately predict CTRs and CVRs at varying price points. With this capability, platforms can recommend coupons that increase not only the profits for advertisers but also the revenue for the platform, as coupons directly affect both the price and the CTRs and CVRs, which in turn changes the auction outcomes.

In previous studies [10, 16], advertisers submit bids once, and the auction mechanism simultaneously determines coupon allocations, ad placements, and payments. To maintain the incentive compatibility property in these settings, complex payment functions are required, which introduces significant implementation challenges. From a practical standpoint, such complexity can hinder the adoption of the mechanism.

In this paper, we address this issue by modeling the auction process as an extensive-form game. The mechanism is split into two stages. In the first stage, advertisers report their coupon bids to get personalized coupon recommendations. In the second stage, advertisers report their auction bids to participate in auctions.

By decoupling the coupon recommendation stage from the auction process, there is no need to design a complicated payment function to ensure incentive compatibility. This is because the coupons are determined in the first stage, which already pin down the CTRs and CVRs. Therefore, in the auction stage, them can still be regarded as constants, as in standard auction settings.

Our goal, in this paper, is to design a coupon recommendation strategy within truthful mechanisms that is easy to implement with desirable economic properties. In particular, our mechanism should be compatible with the prevalent industry standards, i.e., the second-price auctions. Such compatibility ensures that the integration of coupons does not change the rules of the existing auction frameworks.

We summarize our contributions as follows:

- We model the coupon recommendation within truthful auctions as an extensive-form game and provide a sufficient condition in which the advertisers report truthfully in the recommendation stage;
- We construct two mechanisms, which create win-win situations, benefiting the advertisers, the platform, and the users;
- We conduct extensive experiments with both synthetic and industrial datasets to demonstrate the performance of our mechanism.

## 1.1 Related Work

Our work is related to the design of revenue-optimal mechanisms under different settings [10, 12, 15, 18, 24]. There is also a line of work that aims to improve revenue for simple auction mechanisms such as the second-price auction. Different approaches has been proposed include setting appropriate reserve prices [2, 9],boosting the bidders' bids [5, 7], offering coupons to bidders [19, 20] and users [10, 16].

Ni et al. [16] and Liu et al. [10] both consider bid-dependent coupons. The mechanism proposed by Ni et al. [16] is designed for an auto-bidding setting, while the mechanism provided by Liu et al. [10] suffers from implementation issues. Our work also considers the optimization of the platform's revenue by distributing user coupons. The difference is that we focus more on practical applications and aim to design mechanisms that are easy to understand and implement while exhibiting nice economic properties at the same time.

## 2 Preliminaries

We consider a scenario where one seller auctions off a single advertising slot to $n$ advertisers, denoted by $[n] = \{1, 2, \cdots, n\}$. For each advertiser $i$, his value $v_i \in \mathcal{V} = [0, \bar{v}]$ for the slot is drawn from distribution $F_i(v_i)$, with the corresponding density function $f_i(v_i)$. We assume that all advertisers' value distributions are independent and publicly known to both the advertisers and the seller. However, the realized values are the private information of each advertiser.

Unlike the standard auction setting where the CTRs and the CVRs are treated as constants, we consider a new auction scenario in which the CTRs and CVRs are variables, and can be influenced by offering coupons to users. Let $c_i \in C \subseteq \mathbf{R}_+$ denote the coupon offered to the user when advertiser $i$'s ad is displayed.

Let $h_i(c_i)$ be the CTR (or CVR) and of the advertiser $i$'s ad when offering coupon $c_i$ to the user. We assume that the seller employs a truthful auction mechanism, such as second-price auctions or Myerson auctions, to determine the allocation and the payment.

Different from previous work [10], where the platform offers coupons directly to users, in our setting, the platform can only recommend that advertisers offer coupons to users. We model this scenario as an extensive-form game with the following steps:

(1) Advertiser $i$ reports a bid $b_i^c$ to the platform, and the platform recommends a coupon $\phi_i(b_i^c)$ to be offered to the user.
(2) After receiving the coupon recommendation, advertiser $i$ determines whether to follow the platform's recommendation. If advertiser $i$ follows the recommendation, the platform sets $c_i = \phi_i(b_i^c)$, otherwise, $c_i = 0$.
(3) Advertiser $i \in [n]$ reports a bid $b_i^a$ to the platform in the auction process.
(4) The platform determines the allocation and payment based on the auction mechanism.

Formally, we define the mechanism as follows.

DEFINITION 1. *A mechanism $\mathcal{M}$ is a tuple $(x, p, \phi)$, where*

- $\phi = (\phi_1, \phi_2, \cdots, \phi_n)$ *is the coupon function, where $\phi_i$ maps advertiser $i$'s bid $b_i^c$ to coupon $c_i$;*
- $x = (x_1, x_2, \cdots, x_n)$ *is the allocation rule, in which $x_i$ maps the advertisers' bids $b^a$ to the probability of advertiser $i$ winning the auction given advertisers' coupons $c$;*
- $p = (p_1, p_2, \cdots, p_n)$ *is the payment rule, in which $p_i$ maps the advertisers' bids $b^a$ to the payment of advertiser $i$ in the auction given advertisers' coupons $c$.*

We use $v_{-i} = (v_1, \cdots, v_{i-1}, v_{i+1}, \cdots, v_n)$ and $b_{-i}^a = (b_1^a, \cdots, b_{i-1}^a, b_{i+1}^a, \cdots, b_n^a)$ to denote the advertisers' valuation profile and bid profile excluding advertiser $i$, respectively.

Assume that the advertisers have quasi-linear utilities. Formally, for advertiser $i$, if his valuation is $v_i$ and he offers coupon $c_i$ to the user, his utility is

$$u_i(v_i, v_{-i}; c_i) = h_i(c_i)(v_i - c_i)x_i(v_i, v_{-i}; c_i) - p_i(v_i, v_{-i}; c_i).$$

The platform's revenue is $\text{Rev} = \sum_{i=1}^n p_i(v_i, v_{-i}; c_i)$.

We analyze this extensive-form game using backward induction. We first analyze the advertisers' optimal auction bidding strategy in the auction process, then we give a sufficient condition in which the advertisers' optimal bidding strategy in the coupon recommendation process is bidding truthfully.

We assume that the platform uses a truthful mechanism that does not incorporate coupons. As mentioned in the previous section, we aim to integrate the coupon recommendation stage into the mechanism without affecting the existing auction rules. It is known that any truthful auction mechanism $(x, p)$ must satisfy the following Myerson Lemma [15].

LEMMA 1 (MYERSON LEMMA [15]). *The auction mechanism $(x, p)$ is incentive-compatible if and only if*

- $x_i(v_i, v_{-i})$ is monotone non-decreasing with respect to $v_i$.
- $p_i(v_i, v_{-i})$ is uniquely determined by the following function:

$$p_i(v_i, v_{-i}) = v_i x_i(v_i, v_{-i}) - \int_0^{v_i} x_i(s, v_{-i}) ds.$$

Note that in the above lemma, both $x_i$ and $p_i$ are not functions of $c$.

The following lemma shows that advertiser $i$'s optimal bidding strategy with coupon $c_i$ is to bid $v_i - c_i$[1].

LEMMA 2. *For advertiser $i$, his optimal bidding strategy is to report $b_i^a = v_i - c_i$ when the coupon recommendation is $c_i$.*

PROOF. Suppose advertiser $i$ has a valuation $v_i$ and receives a coupon recommendation $c_i$. If advertiser $i$ reports a bid $b_i^a = v_i - c_i$, the utility is given by

$$u_i(v_i, b_{-i}^a; c_i) = h_i(c_i)(v_i - c_i)x_i(v_i - c_i, b_{-i}^a) - p_i(v_i - c_i, b_{-i}^a)$$
$$= h_i(c_i) \int_0^{v_i - c_i} x_i(s, b_{-i}^a) ds,$$

where $x_i(b_i^a, b_{-i}^a)$ is the allocation probability and $p_i(b_i^a, b_{-i}^a)$ is the payment determined by the auction mechanism.

Now, consider a deviation where advertiser $i$ reports a bid $b_i^{a'} = v_i' - c_i$, where $v_i' \neq v_i$. The utility under this deviation becomes

$$u_i(v_i', v_{-i}; c_i) = h_i(c_i)(v_i - c_i)x_i(v_i' - c_i, b_{-i}) - p_i(v_i' - c_i, b_{-i})$$
$$= h_i(c_i)(v_i - v_i')x_i(v_i' - c_i, b_{-i}) + h_i(c_i) \int_0^{v_i' - c_i} x_i(s, b_{-i}) ds$$

Thus, advertiser $i$'s utility gain when reporting $v_i' - c_i$ is

$$\Delta = h_i(c_i)(v_i - v_i')x_i(v_i' - c_i, b_{-i}) + h_i(c_i) \int_{v_i - c_i}^{v_i' - c_i} x_i(s, b_{-i}) ds$$
$$= h_i(c_i)(v_i - v_i')[x_i(v_i' - c_i, b_{-i}) - x_i(\xi, b_{-i})],$$

where $\xi$ lies in the interval between $v_i - c_i$ and $v_i' - c_i$.

Now considering $v_i' < v_i$, it follows that $v_i' - c_i < v_i - c_i$, $x_i(v_i' - c_i, b_{-i}) - x_i(\xi, b_{-i}) < 0$ and $v_i - v_i' > 0$, meaning that $\Delta < 0$.

Following the same analysis, we can also obtain that $\Delta < 0$ when $v_i' > v_i$, which completes the proof. □

Lemma 2 clearly shows that with coupons, mechanism $(x, p)$ is no longer truthful. However, according to the Revelation Principal [14], to make the mechanism truthful, the platform can directly use $v_i - c_i$ to determine both the allocation and payment. In the following analysis, we use $x_i(v_i, v_{-i}; c_i)$ as advertiser $i$'s allocation when offering coupon $c_i$ to users and $p_i(v_i, v_{-i}; c_i)$ as the corresponding payment.

Now, we identify a sufficient condition under which no advertiser can achieve a higher utility by misreporting in the first stage.

LEMMA 3. *Given that other advertisers bid truthfully in the first stage, advertiser $i$ cannot get a higher utility by misreporting if the following two conditions hold.*

- *The coupon function $\phi_i(b_i^c)$ is monotone non-decreasing with respect to $b_i^c$.*
- *The allocation rule is maximized when reporting $b_i^c = v_i$, i.e., $v_i = \arg\max_{b_i^c} x_i(v_i, v_{-i}; \phi_i(b_i^c))$.*

---

[1]In the second stage, the CTRs and the CVRs are pinned down, and the auction process is the same as standard ones.

PROOF. Assume that advertiser $i$ has a true valuation $v_i$. Let $b_i^c = v_i$ denote the truthful bid in the first stage and let $\hat{b}_i^c$ represent any misreported bid. When $\hat{b}_i^c > v_i$, the platform can recommend a coupon $\phi_i(\hat{b}_i^c) > v_i$, resulting in a negative utility for advertiser $i$. Thus, in the following proof, we only need to show advertiser $i$ cannot bid $\hat{b}_i^c$ to get a higher utility.

**Truthful Reporting:** When advertiser $i$ bids truthfully ($b_i^c = v_i$), the platform recommends a coupon $\phi_i(b_i^c) = \phi_i(v_i)$. According to Lemma 2, advertiser $i$ then bids $v_i$ in the auction. The utility for advertiser $i$ in this scenario is

$$u_i(b_i^c) = h_i(\phi_i(b_i^c))(v_i - \phi_i(b_i^c))x_i(v_i, v_{-i}; \phi_i(b_i^c)) - p_i(v_i, v_{-i}; \phi_i(b_i^c))$$
$$= h_i(\phi_i(v_i)) \int_0^{v_i} x_i(s, v_{-i}; \phi_i(v_i)) ds.$$

**Misreporting:** Suppose advertiser $i$ misreports by selecting $\hat{b}_i^c$. The platform then recommends a coupon $\phi_i(\hat{b}_i^c)$. Again, by Lemma 2, advertiser $i$ bids $v_i$ in the auction. The utility in this case is

$$u_i(\hat{b}_i^c) = h_i(\phi_i(\hat{b}_i^c))(v_i - \phi_i(\hat{b}_i^c))x_i(v_i, v_{-i}; \phi_i(\hat{b}_i^c)) - p_i(v_i, v_{-i}; \phi_i(\hat{b}_i^c))$$
$$= h_i(\phi_i(\hat{b}_i^c)) \int_0^{v_i} x_i(s, v_{-i}; \phi_i(\hat{b}_i^c)) ds.$$

**Comparison of Utilities:** Since the allocation rule is maximized when $b_i^c = v_i$, it follows that

$$v_i = \arg\max_{b_i^c} x_i(v_i, v_{-i}; \phi_i(b_i^c)).$$

Therefore, for any misreport $\hat{b}_i^c$,

$$x_i(v_i, v_{-i}; \phi_i(b_i^c)) \geq x_i(v_i, v_{-i}; \phi_i(\hat{b}_i^c)).$$

Integrating both sides over the interval $[0, v_i]$ yields

$$\int_0^{v_i} x_i(s, v_{-i}; \phi_i(b_i^c)) ds \geq \int_0^{v_i} x_i(s, v_{-i}; \phi_i(\hat{b}_i^c)) ds.$$

Additionally, given that the coupon function $\phi_i(b_i^c)$ is monotone non-decreasing and the function $h_i(c_i)$ is also monotone non-decreasing, we have

$$\phi_i(v_i) \geq \phi_i(\hat{b}_i^c) \quad \text{and} \quad h_i(\phi_i(v_i)) \geq h_i(\phi_i(\hat{b}_i^c)).$$

It follows that

$$u_i(b_i^c) = h_i(\phi_i(v_i)) \int_0^{v_i} x_i(s, v_{-i}; \phi_i(v_i)) ds$$
$$\geq h_i(\phi_i(\hat{b}_i^c)) \int_0^{v_i} x_i(s, v_{-i}; \phi_i(\hat{b}_i^c)) ds = u_i(\hat{b}_i^c).$$

Therefore, advertiser $i$ does not gain any additional utility by misreporting his bid. □

Lemma 3 establishes sufficient conditions for the coupon function $\phi_i(v_i)$ that ensure advertiser $i$ bids truthfully in the first stage. Complementing this, Lemma 2 characterizes the advertisers' optimal strategy in the auction stage. Building upon these foundational results, we proceed in the next section to construct the function $\phi_i(v_i)$ and analyze its implications on both the advertisers' utility and the platform's revenue. Furthermore, we demonstrate that adhering to the platform's coupon recommendations yields superior outcomes compared to the strategy of offering no coupons to users.

# 3 Distribution-free auctions

In this section, we analyze the setting where the platform employs distribution-free auctions as the base mechanism. We begin by examining advertisers' utility and platform revenue under second-price auctions.

**Definition 2 (Second-price Auction).** *In a second-price auction, the advertiser with the highest rank-score wins the auction and the payment is the second highest rank-score. Formally, define the rank score as the product of CTRs and the corresponding bids,*

$$r_i(v_i) = h_i(\phi_i(v_i))(v_i - \phi_i(v_i)).$$

*The allocation rule is*

$$x_i(v_i, v_{-i}; \phi_i(v_i)) = \begin{cases} 1 & if \ r_i(v_i) \ge r_j(v_j), \forall j \\ 0 & otherwise \end{cases}, \quad (1)$$

*and the payment rule is*

$$p_i(v_i, v_{-i}; \phi_i(v_i)) = \begin{cases} \max_{j \ne i} r_j(v_j) & if \ r_i(v_i) \ge r_j(v_j), \forall j \\ 0 & otherwise \end{cases}. \quad (2)$$

Let $\Phi_i(v_i)$ be the set of the coupons maximizing the advertiser $i$'s rank-score when valuation is $v_i$:

$$\Phi_i(v_i) = \left\{ c \middle| c \in \arg\max_b h_i(b)(v_i - b) \right\}. \quad (3)$$

Then we define the coupon function as

$$\phi_i(v_i) = \min \{\Phi_i(v_i)\}. \quad (4)$$

Before we dive into our analysis, we introduce the following definition.

**Definition 3 (Increasing Difference).** *A function $g : \mathcal{A} \times \mathcal{B} \mapsto \mathbb{R}$ has increasing difference if for any $(a, b), (a', b') \in \mathcal{A} \times \mathcal{B}$ with $a > a', b > b'$, the following holds:*

$$g(a, b) - g(a, b') \ge g(a', b) - g(a', b').$$

Recall that $h_i(c)$ is a monotone increasing and concave function. Thus, $h_i(c)(v_i - c)$ has increasing difference. Then, we can get the following result.

**Lemma 4.** *The mechanism $\mathcal{M} = (\phi, x, p)$, in which the coupon function is defined in (4), the allocation rule is defined in (1) and the payment rule is defined in (2), ensures that truthfully reporting valuations in both the first and second stages is the optimal strategy for advertisers.*

**Proof.** According to Lemma 2, we immediately conclude that truthfully reporting their valuations is an optimal strategy for advertisers in the second stage, as second-price auctions satisfy the Myerson lemma.

According to Topki's Theorem [21], we know that $\phi_i(v_i)$ is a monotone non-decreasing function.

Based on the definition of the coupon function, we know that

$$h_i(\phi_i(v_i))(v_i - \phi_i(v_i)) \ge h_i(\phi_i(v_i'))(v_i - \phi_i(v_i')),$$

meaning that $v_i \in \arg\max_{b_i^c} x_i(v_i, v_{-i}; \phi_i(b_i^c))$. Applying Lemma 3, we can conclude that truthfully reporting valuations is also an optimal strategy for advertisers in the first stage. □

Now we compare the advertisers' utility and the platform's revenue between this mechanism and the vanilla second-price auctions without coupons.

**Lemma 5.** *Following the platform's recommendation is a weakly dominant strategy for advertisers, and offering coupons to users can weakly increase advertisers' utilities.*

**Proof.** In a distribution-free auction mechanism, one advertiser's payment depends solely on other advertisers' bid profiles. For convenience, we use $r_j(v_j)$ as the highest rank-score other than that of advertiser $i$.

We compare the advertiser $i$'s utility between offering coupons and not offering coupons.

- If advertiser $i$ doesn't choose to offer coupons, his utility is

$$u_i = [h_i(0)v_i - r_j(v_j)]x_i(v_i, v_{-i}; 0).$$

- If advertiser $i$ follows the platform's recommendation and offers coupon $\phi_i(v_i)$ to users, his utility is

$$u_i' = [h_i(\phi_i(v_i))(v_i - \phi_i(v_i)) - r_j(v_j)]x_i(v_i, v_{-i}; \phi_i(v_i)).$$

We know that

$$\phi_i(v_i) = \arg\max_b h_i(b)(v_i - b),$$

which implies

$$h_i(\phi_i(v_i))(v_i - \phi_i(v_i)) \ge h_i(0)v_i.$$

Additionally, since the rank-score is defined as $r_i(v_i) = h_i(\phi_i(v_i))(v_i - \phi_i(v_i))$. We can conclude that $x_i(v_i, v_{-i}; \phi_i(v_i)) \ge x_i(v_i; v_{-i}; 0)$.

Combining these two arguments, we conclude that $u_i' \ge u_i$. Thus, following the platform's recommendation can weakly increase advertiser $i$'s utility. □

**Lemma 6.** *This mechanism weakly increases the platform's revenue.*

**Proof.** For a single auction, we use $r_j(v_j)$ to denote the second-highest rank-score, which also represents the platform's revenue in this auction. According to Lemma 5, we know that following the recommendation to offer coupons is the advertiser $j$'s optimal strategy. Since offering coupons to users weakly increases the second-highest rank score. Thus, this mechanism weakly increases the platform's revenue. □

Combining Lemma 5 and Lemma 6, we immediately obtain the following result.

**Theorem 1.** *Second-price auctions with user coupons create a win-win situation, benefiting the advertisers, the platform, and the users.*

**Proof.** According to Lemma 5 and Lemma 6, second-price auctions with user coupons increases the advertisers' utilities and the platform's revenue.

As for the users, they can purchase products at a lower price with coupons, which is clearly beneficial. □

After analyzing the coupon function in second-price auctions, we then proceed to construct a coupon function in distribution-dependent auctions such as Myerson auctions. Further, we show that Myerson auctions with coupons maximizes revenue. This mechanism simplifies the implementation of the mechanism proposed by Liu et al. [10] but has the same outcome. In equilibrium, these two games are revenue equivalent.

## 4 Distribution-dependent auctions

In this section, we assume that the valuation of the $n$ advertisers are independent random variables. The joint distribution $f(v) : \mathcal{V}^n \mapsto \mathbb{R}$ for the valuation profile $v = (v_1, \cdots, v_n)$ is the product of each advertiser's valuation distribution:

$$f(v) = \prod_{i \in [n]} f_i(v_i).$$

We also define $f_{-i}(v_{-i}) : \mathcal{V}^{n-1} \mapsto \mathbb{R}$ for the valuation profile $v_{-i}$ as

$$f_{-i}(v_{-i}) = \prod_{j \in [n], j \neq i} f_j(v_j)$$

Following a similar induction process as in Myerson [15], we first derive the virtual value function for this case, which will be helpful in constructing a coupon function.

LEMMA 7. *Aftering fixing each advertiser's coupon recommendation $c = (c_1, \cdots, c_n)$, the platform's revenue can be expressed as*

$$Rev = \int_v \left[ \sum_{i=1}^n h_i(c_i) \left( v_i - c_i - \frac{1 - F_i(v)}{f_i(v)} \right) x_i(v_i, v_{-i}; c_i) \right] f(v) \, dv$$

PROOF. According to Lemma 2, we know that in the auction stage, the advertisers' optimal bidding strategy is to bid their valuation profile truthfully.

Following the Myerson Lemma [15], we know

$$Rev = \int_v \left[ \sum_{i=1}^n p_i(v_i, v_{-i}) \right] f(v) \, dv$$

$$= \int_v \left[ \sum_{i=1}^n h_i(c_i)(v_i - c_i) x_i(v_i, v_{-i}; c_i) \right.$$

$$\left. - h_i(c_i) \int_0^{v_i} x_i(s, v_{-i}; c_i) \, ds \right] f(v) \, dv$$

$$= \int_{v_{-i}} \left[ \sum_{i=1}^n \int_{v_i} h_i(c_i)(v_i - c_i) x_i(v_i, v_{-i}; c_i) f_i(v_i) \, dv_i \right.$$

$$\left. - h_i(c_i) \int_{v_i} \int_0^{v_i} x_i(s, v_{-i}; c_i) f_i(v_i) \, ds dv_i \right] f_{-i}(v_{-i}) \, dv_{-i}$$

By integrating by parts, we have

$$Rev = \int_{v_{-i}} \left[ \sum_{i=1}^n h_i(c_i) \right.$$

$$\int_{v_i} \left( v_i - c_i - \frac{1 - F_i(v_i)}{f_i(v_i)} \right) x_i(v_i, v_{-i}; c_i) f_i(v_i) \, dv_i \right] f_{-i}(v_{-i}) \, dv_{-i}$$

$$= \int_v \left[ \sum_{i=1}^n h_i(c_i) \left( v_i - c_i - \frac{1 - F_i(v_i)}{f_i(v_i)} \right) x_i(v_i, v_{-i}; c_i) \right] f(v) \, dv$$

□

We now define the auction mechanism to be analyzed in this section. Let the advertiser $i$'s rank score be

$$r_i(v_i; c_i) = h_i(c_i) \left( v_i - c_i - \frac{1 - F_i(v_i)}{f_i(v_i)} \right).$$

The allocation rule is defined as follows:

$$x_i(v_i, v_{-i}; c_i) = \begin{cases} 1 & \text{if } r_i(v_i; c_i) \geq r_j(v_j; c_j), \forall j \\ 0 & \text{otherwise} \end{cases}, \quad (5)$$

and the payment rule is:

$$p_i(v_i, v_{-i}; c_i)$$

$$= h_i(c_i) \left[ (v_i - c_i) x_i(v_i, v_{-i}; c_i) - \int_0^{v_i} x_i(s, v_{-i}; c_i) \, ds \right]$$

$$= \begin{cases} h_i(c_i)(w_i - c_i) & \text{if } r_i(v_i; c_i) \geq r_j(v_j; c_j), \forall j \\ 0 & \text{otherwise} \end{cases}, \quad (6)$$

where $w_i$ is the minimum bid ensuring advertiser $i$ wins the auction, i.e.,

$$h_i(c_i) \left( w_i - c_i - \frac{1 - F_i(w_i)}{f_i(w_i)} \right) = \max_{j \neq i} r_j(v_j; c_j).$$

Similarly, define the coupons maximizing the advertise $i$'s rank-score when valuation is $v_i$ as the set $\Phi_i(v_i)$

$$\Phi_i(v_i) = \left\{ c \,\middle|\, c \in \arg\max_b h_i(b) \left( v_i - b - \frac{1 - F_i(v_i)}{f_i(v_i)} \right) \right\},$$

and the coupon function as

$$\phi_i(v_i) = \min\{\Phi_i(v_i)\}. \quad (7)$$

In the following analysis, we assume the virtual value function $r_i(v_i; c_i)$ is monotone non-decreasing with respect to $v_i$. In the general setting, if $r_i(v_i; c_i)$ is not monotone non-decreasing with respect to $v_i$, we can use the so-called "ironing" technique to obtain an ironed virtual value function $\bar{r}_i(v_i; c_i)$ which is monotone non-decreasing without losing any revenue.

LEMMA 8. *In the mechanism $\mathcal{M} = (\phi, x, p)$, where the coupon function, the allocation rule and the payment rule are defined in (7), (5) and (6), respectively, truthfully reporting their valuation in both the first and second stages is optimal for advertisers.*

PROOF. First, $x_i(v_i, v_{-i}; c_i)$ is monotone non-decreasing with respect to $v_i$ as $v_i - \frac{1 - F_i(v_i)}{f_i(v_i)}$ is monotone non-decreasing with respect to $v_i$.

According to the Myerson lemma [15], the payment rule is identical defined as

$$p_i(v_i, v_{-i}; c_i) = h_i(c_i) \left[ (v_i - c_i) x_i(v_i, v_{-i}; c_i) - \int_0^{v_i} x_i(s, v_{-i}; c_i) \, ds \right]$$

$$= \begin{cases} h_i(c_i)(w_i - c_i) & \text{if } r_i(v_i; c_i) \geq r_j(v_j; c_j), \forall j \\ 0 & \text{otherwise} \end{cases},$$

where $w_i$ is the minimum value ensuring advertiser $i$ wins the auction, i.e.,

$$h_i(c_i) \left( w_i - c_i - \frac{1 - F_i(w_i)}{f_i(w_i)} \right) = \max_{j \neq i} r_j(v_j; c_j).$$

In the auction stage, since the advertisers' bids do not affect the coupon offered to users, it is an optimal strategy for advertisers bidding truthfully in the auction stage.

Next, we show that it is also an optimal strategy for advertisers to report their true valuation in the coupon recommendation stage.

Since $v_i - \frac{1-F_i(v_i)}{f_i(v_i)}$ is monotone non-decreasing with respect to $v_i$ and $h_i(c_i)$ is monotone non-decreasing with respect to $c_i$. Thus we know the following function

$$h_i(c_i)\left[v_i - c_i - \frac{1-F_i(v_i)}{f_i(v_i)}\right]$$

has increasing differences. According to Topki's Theorem [21], we know $\phi_i(v_i)$ is also non-decreasing with respect to $v_i$.

Based on the definition of the coupon function in (7), we have that

$$h_i(\phi_i(v_i))\left(v_i - \frac{1-F_i(v_i)}{f_i(v_i)} - \phi_i(v_i)\right)$$
$$\geq h_i(\phi_i(v_i'))\left(v_i - \frac{1-F_i(v_i)}{f_i(v_i)} - \phi_i(v_i')\right),$$

making truthfully reporting an optimal strategy in the coupon recommendation stage based on Lemma 3. □

After showing the advertisers' optimal strategy in both the first and the second stages, we now consider the platform's optimal revenue in our setting.

Lemma 7 shows the platform's revenue expressed by the virtual value function, which is the same as the findings of Liu et al. [10]. Their work constructs a revenue-optimal mechanism for the platform. Consequently, we can conclude that our mechanism also achieves optimal revenue. However, it's important to note that their mechanism has a complicated integral computation process, but our method avoids it.

We will now proceed to demonstrate that our mechanism increases both the advertisers' utilities and the platform's revenue.

Lemma 9. *Following the platform's recommendation is a weakly dominant strategy for advertisers and offering coupons to users can weakly increase advertiser's utilities.*

Proof. According to Lemma 7, in one auction, the advertiser's payment can be written as

$$p_i(v_i, v_{-i}; \phi_i(v_i)) = h_i(\phi_i(v_i))\left[v_i - \frac{1-F_i(v_i)}{f_i(v_i)} - \phi_i(v_i)\right]$$
$$\cdot x_i(v_i, v_{-i}; \phi_i(v_i)).$$

If advertiser $i$ follows the platform's recommendation, advertiser $i$'s utility is

$$u_i = h_i(\phi_i(v_i))\left[\frac{1-F_i(v_i)}{f_i(v_i)}\right]x_i(v_i, v_{-i}; \phi_i(v_i)).$$

If advertiser $i$ doesn't follow the platform's recommendation, advertiser $i$'s utility is

$$u_i' = h_i(0)\left[v_i - \frac{1-F_i(v_i)}{f_i(v_i)}\right]x_i(v_i, v_{-i}; 0).$$

Since $h_i(\phi_i(v_i)) \geq h_i(0)$ and $x_i(v_i, v_{-i}; \phi_i(v_i)) \geq x_i(v_i, v_{-i}; 0)$, it follows that $u_i \geq u_i'$, making it optimal for advertiser $i$ to follow the platform's recommendation. □

Then we show that this mechanism also increases the platform's revenue.

Lemma 10. *Offering coupons to users weakly increases the platform's revenue in Myerson auctions.*

Proof. If offering coupons to users, we know the platform's revenue is

$$\text{Rev} = \int_v\left[\sum_{i=1}^n h_i(c_i)\left(v_i - c_i - \frac{1-F(v)}{f(v)}\right)x_i(v_i, v_{-i}; c_i)\right]f(v)\,\mathrm{d}v.$$

If the platform doesn't offer coupons to users, the revenue is

$$\text{Rev}' = \int_v\left[\sum_{i=1}^n h_i(0)\left(v_i - \frac{1-F(v)}{f(v)}\right)x_i(v_i, v_{-i}; 0)\right]f(v)\,\mathrm{d}v.$$

Based on the coupon function (7), we know that

$$h_i(c_i)\left(v_i - c_i - \frac{1-F_i(v_i)}{f_i(v_i)}\right) \geq h_i(0)\left(v_i - \frac{1-F_i(v_i)}{f_i(v_i)}\right),$$

and

$$x_i(v_i, v_{-i}; c_i) \geq x_i(v_i, v_{-i}; 0).$$

Each item in *Rev* is weakly larger than the corresponding item in *Rev'*. Thus, we have $Rev \geq Rev'$. □

Then we can further get the following theorem.

Theorem 2. *By offering coupons to users, the mechanism $\mathcal{M} = (\phi, x, p)$, where $\phi$ is defined in (7), $x$ is defined in (5) and $p$ is defined in (6), creates a win-win situation, benefiting the advertisers, the platform and the users. Moreover, this mechanism achieves the optimal revenue for the platform.*

Proof. According to Lemma 9 and Lemma 10, we know that Myerson auctions with user coupons weakly increase the advertisers' utilities and the platform's revenue. The users also have a lower price to buy the product after receiving the coupons. Thus, this mechanism creates a win-win situation.

Furthermore, according to Lemma 6, our coupon function maximizes the virtual value function pointwise. Since the allocation rule allocates the item with the highest virtual value. Thus, this allocation rule optimizes the platform's revenue. So, this mechanism is revenue-optimal. □

We have constructed two mechanisms, one is a second-price auction with user coupons and the other one is Myerson auction with user coupons. The former mechanism doesn't rely on the knowledge of advertisers' value distributions, while the latter relies on the knowledge of advertisers' value distributions. In the next section, we provide experimental results to verify our mechanisms.

## 5 Experiments

We evaluate our proposed mechanisms through a series of experiments using both synthetic and industrial data[2]. We compare the following mechanisms:

(1) Second-price Auctions (SPA);
(2) Myerson Auctions (MA);
(3) Coupon Auctions [10] (CA);
(4) Second-price Auctions with Coupons (SPA-C);
(5) Myerson Auctions with Coupons (MA-C).

[2]The codes are provided in https://anonymous.4open.science/r/coupon_auction65DC

The first three mechanisms (SPA, MA, and CA) serve as baselines and the last two mechanisms (SPA-C, MA-C) are proposed in this paper.

It's important to note that SPA and MA do not offer coupons to users and CA provides coupons based on a complex payment rule designed to satisfy incentive compatibility (IC) conditions. Across all baseline mechanisms and our two proposed mechanisms, the optimal bidding strategy is for users to report their valuations truthfully. One difference between the baseline mechanisms and ours is that the advertisers bid only once in the baseline mechanisms, while they bid twice in our mechanisms.

## 5.1 Synthetic dataset

We generate a synthetic dataset for our experiments following the method described by Ni et al. [16]. The dataset is constructed with the following parameters and assumptions:

(1) Coupon Set: We use a pre-defined set of four possible coupon values: $C = \{0, 2, 4, 8\}$.
(2) We consider there are $n = 8$ advertisers participating in $m = 100,000$ auctions.
(3) Value Distributions: We examine two different value distributions.
   (a) Uniform Distribution $v_i \sim \mathcal{U}[5, 50]$
   (b) Log-normal Distribution: $\ln v_i \sim N(3, 1)$.
(4) CTRs:
   - Base CTR(without coupons): $h_i(0) \sim \mathcal{U}[0.005, 0.5]$;
   - CTR with coupon value 2: $h_i(2) = 1.1h_i(0)$;
   - CTR with coupon value 4: $h_i(4) = 1.2h_i(0)$;
   - CTR with coupon value 8: $h_i(8) = 1.3h_i(0)$;

The CTRs are generated to reflect the impact of different coupon values on user engagement. In each auction instance, the value for each advertiser is independently drawn from the same distribution (either uniform or log-normal). Thus, we generate two synthetic datasets. These two datasets allow us to evaluate our proposed mechanisms (SPA-C and MA-C) against the baseline methods under controlled conditions, enabling a comprehensive analysis of their performance across various scenarios.

After describing the dataset generation method, we now detail the simulation procedure for our proposed mechanisms (SPA-C and MA-C) and the baseline methods.

Our proposed mechanisms are simulated using a two-stage process:

(1) Coupon Recommendation Stage:
   - Receive the advertisers' valuation profile.
   - Enumerate all possible coupons.
   - Determine the optimal coupons to recommend to each advertiser.
(2) Auction Stage:
   - Determine the corresponding CTRs based on the recommended coupons.
   - Receive the advertisers' valuation profile as bids.
   - Apply the corresponding allocation rule and payment rule to determine the final allocation and payment.

The baseline methods are simulated by bidding once.

(1) Receive the advertisers' valuation profile once.

(2) Simultaneously determine the coupon, allocation, and payment.

We now present our experimental results to evaluate the effectiveness of our proposed methods. Our first objective is to demonstrate that following the platform's coupon recommendations leads to better utilities for advertisers.

To illustrate the advantage of following the platform's recommendations, we conducted the following experiment:

(1) We randomly selected one advertiser, denoted as advertiser $k$.
(2) All other advertisers are set to follow the platform's coupon recommendations.
(3) We compared advertiser $k$'s utility under two scenarios:
   (a) Following the platform's recommendation;
   (b) Ignoring the platform's recommendation.

This comparison allows us to isolate the effect of following recommendations on an individual advertiser's utility while keeping the behavior of other advertisers constant. The results are shown in Figure 1. Figure 1(a) presents the experimental results in the uniform distribution dataset and Figure 1(b) shows the experimental results in the lognormal distribution dataset. In both recommendation coupons in second-price auctions and recommendation coupons in Myerson auctions, following the platform's recommendation leads to better utilities for one advertiser. In the uniform distribution dataset, following the platform's recommendation increases 35% in second-price auctions and 17% in Myerson auctions. In the lognormal distribution dataset, following the platform's recommendation increases 38% in second-price auctions and 21% in Myerson auctions.

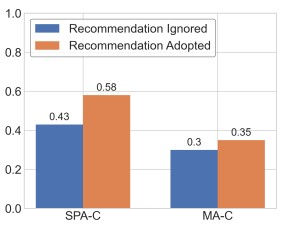
(a) Uniform Distribution

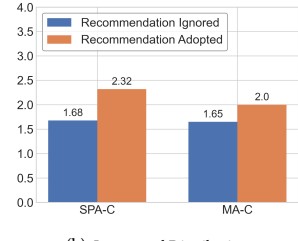
(b) Lognormal Distribution

**Figure 1: The advertiser's utility**

After showing following the platform's recommendation leads to better utility for one advertiser, we then show that our mechanism also increases the platform's revenue. We also show that our mechanism with Myerson auctions has the same outcome compared to the coupon auction proposed by Liu et al. [10].

In Figure 2, we show the advertisers' total utility and the platform's revenue in different mechanisms. In the uniform distribution dataset, Myerson auctions have a higher revenue for the platform compared to second-price auctions, but a lower advertisers' total utilities. The reason is that second-price auctions maximize social welfare while Myerson auctions maximize revenue. Compared to second-price auctions, offering coupons increases the advertisers' total utility by 13% and the platform's revenue by 6%. Offering

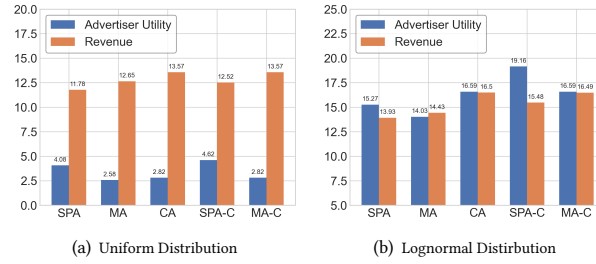

(a) Uniform Distribution      (b) Lognormal Distirbution

**Figure 2: The advertiser's utility and platform's revenue.**

coupons in Myerson auctions increases the advertisers' total utility by 9% and the platform's revenue by 7%. Compared to the coupon auctions, recommending coupons in Myerson auctions have the same outcome, which is consistent with our theoretical results.

In the lognormal distribution dataset, offering coupons increases the advertisers' total utility by 25% and the platform's revenue by 11% in second-price auctions. In Myerson auctions, offering coupons increases the advertisers' total utility by 18% and the platform's revenue by 14%. Experimental results show that although Myerson auctions with user coupons have higher revenue than second-price auctions with user coupons. However, second-price auctions with user coupons have higher advertisers' total utility but sacrifice a small fraction of revenue, which also shows the superiority of distribution-free auctions.

## 5.2 Industrial dataset

To validate our proposed methods in a real-world context, we conducted experiments using an industrial dataset obtained from a major short-form video and live-streaming platform. Since the platform charges the advertisers in a per-impression fashion. Thus, we modified our mechanism, in which the $h_i(\cdot)$ represents the CVR of advertiser $i$'s ad for one user. Since the platform runs second-price auctions to sell ad slots. It is well known that second-price auctions are incentive-compatible. Thus, the advertisers' historical bids are their values.

We collected data over a period of 14 days, selecting 1,000 ads across diverse product categories. The dataset contains rich feature sets: user features (e.g., district, gender), product features (e.g., price, category), and ad features (e.g., format, placement). Coupon values, ranging from 5 to 20, were randomly assigned to users during the data collection period. Then we use this dataset to train a CVR prediction model to predict the ad's CVRs under different users and different coupon values. For our auction simulation, we selected the top 100 advertisers based on ad spend, ensuring a representative sample of the platform's advertising ecosystem.

We select 100,000 auction logs to simulate the auction process. Similar to the experiments in the synthetic dataset, we conduct experiments using the following steps.

(1) Coupon Recommendation Stage:
- Receive the advertisers' valuation profile;
- Discrete the coupon space for each advertiser, and enumerate all possible coupons

- Determine the optimal coupons to recommend to each advertiser.
(2) Auction Stage:
- Based on recommended coupons to determine each advertiser's corresponding CVRs;
- Receive the advertisers' valuation profile as bids;
- Apply the corresponding allocation rule and payment rule to determine the final allocation and payment.

For the coupon auction, we use the same methods to discrete the coupon space and use the corresponding coupon function, allocation rule, and payment rule to determine the coupon, the allocation, and the payment simultaneously.

We now present our experimental results to show the effectiveness of our proposed methods. In the industrial dataset, one difference is that each advertiser's value is independently drawn from a different value distribution compared to the synthetic datasets. In Figure 3, we show that following the platform's recommendation leads to higher utility for each advertiser in both second-price auctions with user coupons and Myerson auctions with user coupons. In second-price auctions with user coupons, following the platform's recommendation increases one advertiser's utility by 72%. In Myerson auctions with user coupons, following the platform's recommendation increases one advertiser's utility by 65%.

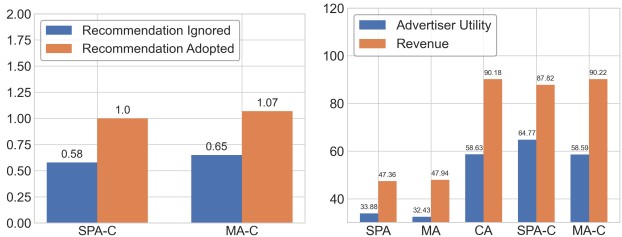

**Figure 3: Advertiser's utility**    **Figure 4: Advertisers' utility and platform's revenue**

In Figure 4, we compare the platform's revenue and the advertisers' total utility in different mechanisms. In second-price auctions, offering coupons increases the advertisers' total utility by 91% and the platform's revenue by 85%. In Myerson auctions, offering coupons increases the advertisers' total utility by 81% and the platform's revenue by 88%. Compared to coupon auctions proposed by Liu et al. [10], Myerson auctions with user coupons have the same outcome when ignoring the precision of numerical calculations.

## 6 Conclusion

In this paper, we addressed the challenge of designing an optimal coupon recommendation strategy within truthful auction mechanisms. Our research has led to the development of two novel mechanisms: a mechanism that incorporates user coupons into second-price auctions, which can be readily integrated into existing auction systems, and a mechanism that incorporates user coupons into Myerson auctions, achieving optimal revenue for the platform. Both theoretical and experimental analyses show that our mechanisms create win-win situations for all the participating parties.

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
