# OpenReview forum: "Optimizing Revenue through User Coupon Recommendations in Truthful Online Ad Auctions"
_ACM.org/TheWebConf/2025/Conference — WWW 2025 Oral_

### Official Review · Reviewer_1Lfu · 2024-11-14

**Novelty:** 3
**Technical Quality:** 3

**Review:**

This paper presents a novel approach to increasing revenue in online ad auctions by incorporating user-targeted coupon recommendations. They model the interaction between advertisers and the platform as a game with two stages. The first stage is coupon bidding, and the second one is auction bidding. The authors design mechanisms that encourage truthful reporting from advertisers while boosting click-through rates (CTRs) and conversion rates (CVRs).

This paper proposes two mechanisms, the first one is a distribution-free model compatible with second-price auctions and the second one is an advanced revenue-optimal mechanism that aligns with Myerson auctions while simplifying complex payment calculations required in prior models.

Finally, the authors evaluate their mechanisms through a series of experiments using both synthetic and industrial data. The experimental results indicate that these mechanisms increase advertiser utility and platform revenue, creating a mutually beneficial structure for advertisers, the platform, and end-users through more strategic coupon use.

This paper addresses the intersection of auction theory and consumer-targeted coupon strategies. Instead of treating CTRs and CVRs as constants, the authors consider a new scenario in which CTRs and CVRs are variables. The two-stage game approach that separates coupon and auction bidding is novel in its simplicity of implementation. I believe this is an interesting direction worth exploring.

However, the description of some notations in the article is quite confusing, which makes the analysis hard to follow. For example, when defining the allocation rule x, I think the input should be  b^a but v is used throughout the paper; I also did not see the dependence of the payment function p on the bids. Therefore I cannot fully understand the model.

Typos:

1.	Page 2, Line 126, “them” should be “they”.

2.	Page 2, Line 152, “Different approaches has” should be “Different approaches have”.

3.	Page 2, Line 153, “[2,9],boostiong” missing a space.

4.	Page 2, Line 180, “(or CVR) and of” should be “(or CVR) of”.

5.	Page 2, Line 204, “$b^a$” should be “$b_i^a$”.

6.	Page 2, Line 207, “$b^a$” should be “$b_i^a$”.

7.	Page 2, Line 218, “Rev” should be “$Rev$”.

8.	Page 3, Line 272, “Principal” should be “Principle

9.	Page 5, Line 468, “maximizes” should be “maximize”.

10.	Page 5, Line 489, “Aftering” should be “After”.

11.	Page 6, Line 589, “differences” should be “difference”.

12.	Page 6, Line 615, “advertiser’s utilities” should be "advertisers’ utilities".

**Questions:**

1.	What is the formal definition of the allocation rule $x$? Why use $v_i$ instead of $b_i^a$ in the notation?

2.	What is the formal definition of the payment rule $p$?

3.	In the proof of Lemma 2, why the utility can be simplified as $h_i(c_i) \int_{0}^{v_i – c_i} x_i(s, b_{-i}^a) ds$ ?

**Reviewer Confidence:**

3: The reviewer is confident but not certain that the evaluation is correct

**Scope:**

3: The work is somewhat relevant to the Web and to the track, and is of narrow interest to a sub-community

---

### Official Review · Reviewer_ZzD1 · 2024-11-26

**Novelty:** 5
**Technical Quality:** 5

**Review:**

This paper studies ad auctions with coupons.
Specifically, it divides the auctions into two stages, where every advertiser reports a coupon bid b1 in the first stage, and reports an ad bid b2 in the second.
The platform recommends a coupon price c(b1) given b1, and if accepted, the utility of the advertiser would be roughly CTR(c(b1)) (v-c(b1)) - p, where v and p are the private value and the payment price; otherwise, CTR(0) v - p.

Fixing a coupon price, the second stage can be made truthful by a routine mechanism such as the second price .
The main contribution is discovering a sufficient condition of coupon function c for advertisers to report b1 truthfully.
The authors further show that the coupon function can weakly increases the utility of both the advertiser and the platform, leading to a win-win situation.

They examine the effectiveness of proposed mechanism on both simulated and industrial data, showing that the recommended coupons are useful.

The paper's formulation and approach look sensible to me.
But I have to acknowledge that I am not an expert in mechanism design, so I may not be able to verify the correctness and may miss important ideas.

Some suggestions:
- It is confusing to use v_i in Sec. 3 for coupon functions. Use b^c_i or clarify that it is safe to assume they are equal.
- It is not clear why it is necessary to discuss Sec. 3 and 4 separately. Please summarize beforehand what can be gained when auctions are distribution-dependent, to give readers some clues beforehand.

**Questions:**

N/A

**Reviewer Confidence:**

2: The reviewer is willing to defend the evaluation, but it is likely that the reviewer did not understand parts of the paper

**Scope:**

4: The work is relevant to the Web and to the track, and is of broad interest to the community

---

### Official Review · Reviewer_1QMK · 2024-12-01

**Novelty:** 5
**Technical Quality:** 5

**Review:**

This paper considers a variant of an online advertising problem. The auction consists of two stages: advertisers first report coupon bids to get personalized coupon recommendations. And then, advertisers report their auction bids to participate in auctions. The goal is to design a coupon recommendation strategy within a truthful mechanism such that the total revenue is maximized.

The main contribution is two mechanisms: one is a second price auction with user coupons, and the second one is a Myerson auction with user coupons. I think the paper studies a clear, interesting, and well-motivated problem. I also appreciate that the presentation and organization of the paper are clear. The proposed mechanisms are quite easy, and thus, they might have a positive impact in reality. On the downside, there is not much surprise in techniques. Both mechanisms follow the similar idea of the well-known techniques.

**Questions:**

I don't have any specific questions.

**Reviewer Confidence:**

3: The reviewer is confident but not certain that the evaluation is correct

**Scope:**

4: The work is relevant to the Web and to the track, and is of broad interest to the community

---

### Official Review · Reviewer_s6UX · 2024-12-02

**Novelty:** 4
**Technical Quality:** 5

**Review:**

The authors study the problem of an advertisement auction platform advising bidders to offer coupons to users with the goal of increasing the revenue and welfare of the auction. At its core, the idea is that offering a coupon to a user can increase the clickthrough rate (CTR) or conversion rate (CVR) of an advertisement which thereby can increase revenue or welfare.  On the other hand, offering a discount on the product necessarily decreases the revenue of the item when it is sold.  As such, the platform would like to incentivize bidding advertisers to provide coupons to users when the platform suggests it.  To that end, the authors study the design of mechanisms which simultaneously: (i) incentivize the advertisers to truthfully bid their value for being displayed and (ii) incentivize the advertisers to follow the recommendation (regarding coupons) suggested by the platform.  The authors find a two-stage mechanism where bidders report two bids, one in a coupon recommendation phase and the second in an allocation phase, where truthful bidding in both phases forms an Nash equilibrium (i.e., the end-to-end mechanism is ex-post incentive compatible).

Positives

* The question studied in the paper is natural since targeting coupons to users has the potential to increase both revenue and welfare of the platform, advertisers, and users

* The authors provide a revenue optimal mechanism for the setting with prior distributions as well as a mechanism providing greater ex-post revenue than the second-price auction in prior-free settings.  Moreover, these mechanisms increase the sum of the advertisers’ utilities.

* The “sequential composition” flavor of the mechanisms is convenient, making the analysis straightforward and potentially facilitating easier adoption (since the coupon recommendation phase is somewhat decoupled from the allocation phase)

Negatives

* The novelty and technical contribution of the result is significantly lessened, in my view, in light of the result of Liu et al. [10] (this issue is exacerbated since, as far as I understand, the mechanism in this paper is only ex-post incentive compatible – see, e.g., Lemma 3 – whereas the mechanism of Liu et al. [10] is dominant strategy incentive-compatible)

Typos/wording issues

* Abstract: What is an “industrial system”?  I’d suggest rewording.
* Page 2, lines 131-132: “prevalent industry standards” -> Is the second-price auction the most standard auction form (rather than, e.g., GSP or first-price?  (See question below)
* Page 2: “approaches has been” -> “approaches have been”
* Page 2: “[2,9],boosting” -> “[2,9], boosting”
* Page 2: “suffers from implementation issues” -> What implementation issues?
* Page 4: “Since offering coupons to users weakly increases the second-highest rank score.” -> This is a sentence fragment; I’d suggest rewording.
* Page 5: “Myerson lemma” -> Elsewhere you capitalize “Myerson Lemma”.  I’d suggest using “Myerson’s Lemma” (or “Myerson’s lemma”) consistently.
* Page 6: Is the equation for $u’_i$ correct in the proof of Lemma 9?  I suspect it should not have $v_i$.
* Page 8: “Since the platform runs second-price auctions to sell ad slots.” -> This is a sentence fragment.
* Page 8: “Discrete the” and “to discrete the” -> “Discretize the” and “to discretize the”

**Questions:**

Can your results extend at all to non-IC auctions?  I understand that your goal is to find optimal incentive compatible auctions, but I believe that many major platforms are turning toward non-IC auctions (in particular, variants of the first-price auction).  This is, in part, due to the credibility of the first-price auction (bidders pay precisely what they bid, so there is no need to "trust" the auctioneer/platform).  As such, can you speak to how one could use a similar approach to yours for improving revenue/welfare/utilities using coupon recommendations in first-price auctions?

**Reviewer Confidence:**

3: The reviewer is confident but not certain that the evaluation is correct

**Scope:**

3: The work is somewhat relevant to the Web and to the track, and is of narrow interest to a sub-community

---

### Official Review · Reviewer_yD8Z · 2024-12-03

**Novelty:** 6
**Technical Quality:** 7

**Review:**

### Summary
This paper studies the setting of online ad auctions with coupons: advertisers can increase profits by offering personalized coupons to customers on an online platform. Often the platform has detailed customer data that can help select a coupon to offer, but if the platform selects and offers coupons directly based on the advertiser's bid, the auction must be quite complicated in order to maintain incentive compatibility. In this work the authors propose an alternative framework where advertisers submit bids in two stages: first, the advertisers submit a valuation, and the platform uses the valuation to suggest a coupon. Then, the advertisers decide whether to accept the suggestion, and submit a bid. In this two-stage setting, the authors find simple incentive-compatible mechanisms and show in industrial data that the advertisers and the platform are better off under these mechanisms.

### Strengths
- This paper gives two mechanisms for implementing coupons in ad auctions, and show that these mechanisms satisfy nice properties including incentive compatibility and simplicity.
- The experimental results are compelling and show a substantial improvement for both advertisers and the platform when using coupons.
- The introduction, results, and proofs are generally clearly written- this paper was enjoyable to read. The proofs appear correct.

### Weaknesses
- My understanding is that the key contribution of this paper is that it offers a more implementable coupon mechanism than previous work by Liu et al. (2024). However, the authors do not argue why Liu's mechanism is not implementable, and examining the paper of Liu et al it appears that they do implement their mechanism for the purpose of experiments. This is especially important because the new mechanism (a) involves two stages of bidding and (b) not intended for an auto-bidding setting, so that the authors' mechanism would add complexity to the auction process.
- The authors argue that the mechanism improves user utility as well because coupons can only decrease the price of items. However, this only improves user utility if the advertisers hold prices constant. However, it seems realistic that advertisers would prices and use coupons to engage in price discrimination, thereby harming user utility.
- It is unclear what insights the synthetic experiments provide; they seem to be conducted in exactly the setting of the theoretical results.

### Additional comments
- It would be helpful if the assumptions on $h_i$ (monotone increasing and concave) were introduced earlier in the paper.
- The role of increasing difference in the proof of Lemma 4 is not clear.
- The proof of Lemma 7 is difficult to read as there is a sum over $i$ and an integral over $v_{-i}$, and these two $i$'s are different, making it difficult to evaluate this specific proof.

**Questions:**

(See "Weaknesses" above for more details/context)
1. What makes the mechanism in Liu et al. (2024) difficult to implement, and in particular how should I think about the tradeoff between the complexity of the payment function and the inefficiency introduced by having two stages of bidding?

2. Is it possible that coupons could decrease user utility (by causing price discrimination)?

3. What is the purpose of the synthetic experiments; in particular, what insights do they provide beyond the theoretical results and the industrial experiments?

**Reviewer Confidence:**

3: The reviewer is confident but not certain that the evaluation is correct

**Scope:**

3: The work is somewhat relevant to the Web and to the track, and is of narrow interest to a sub-community